# Implementation of rapid and frequent SARS-CoV2 antigen testing and response in congregate homeless shelters

**Andrés Aranda-Díaz** [1,2ʸ] *, **Elizabeth Imbert** [2,3ʸ], **Sarah Strieff** [4], **Dave Graham-Squire** [3,5], **Jennifer L. Evans** [3,5], **Jamie Moore** [4], **Willi McFarland** [4,6], **Jonathan Fuchs** [4], **Margaret A. Handley** [3,5,6,7], **Margot Kushel** [3,5]

1 Department of Bioengineering, Stanford University, Stanford, California, United States of America, 2 Division of HIV, ID and Global Medicine, University of California, San Francisco, California, United States of America, 3 UCSF Benioff Homelessness and Housing Initiative, San Francisco, California, United States of America, 4 San Francisco Department of Public Health, San Francisco, California, United States of America, 5 UCSF Center for Vulnerable Populations, University of California, San Francisco, California, United States of America, 6 Department of Epidemiology and Biostatistics, University of California, San Francisco, California, United States of America, 7 UCSF PRISE Center: Partnerships for Research in Implementation Science for Equity, San Francisco, California, United States of America

ʸ These authors contributed equally to this work.
* andres.arandadiaz@ucsf.edu

**Data Availability Statement:** All relevant data are within the manuscript and its Supporting information files.

## Abstract

### Background

People experiencing homelessness who live in congregate shelters are at high risk of SARS-CoV2 transmission and severe COVID-19. Current screening and response protocols using rRT-PCR in homeless shelters are expensive, require specialized staff and have delays in returning results and implementing responses.

### Methods

We piloted a program to offer frequent, rapid antigen-based tests (BinaxNOW) to residents and staff of congregate-living shelters in San Francisco, California, from January 15th to February 19th, 2021. We used the Reach-Effectiveness-Adoption-Implementation-Maintenance (RE-AIM) framework to evaluate the implementation.

### Results

Reach: We offered testing at ten of twelve eligible shelters. Shelter residents and staff had variable participation across shelters; approximately half of eligible individuals tested at least once; few tested consistently during the study.

Effectiveness: 2.2% of participants tested positive. We identified three outbreaks, but none exceeded 5 cases. All BinaxNOW-positive participants were isolated or left the shelters.

Adoption: We offered testing to all eligible participants within weeks of the project's initiation.

**Funding:** The project received funding from the UCSF Benioff Homelessness and Housing Initiative, Kaiser Community Benefits, and Heluna Health. The funders had no role in study design, data collection and analysis, decision to publish, or preparation of the manuscript.

Implementation: Adaptations made to increase reach and improve consistency were promptly implemented.

Maintenance: San Francisco Department of Public Health expanded and maintained testing with minimal support after the end of the pilot.

## Conclusion

Rapid and frequent antigen testing for SARS-CoV2 in homeless shelters is a viable alternative to rRT-PCR testing that can lead to immediate isolation of infectious individuals. Using the RE-AIM framework, we evaluated and adapted interventions to enable the expansion and maintenance of protocols.

## Introduction

People experiencing homelessness are at higher risk of severe COVID-19 due to their older age and comorbidities [1–5]. Congregate shelters, where the majority of people experiencing homelessness in the United States stay [6], present challenges to controlling airborne diseases, including crowding and transient populations [7–9].

COVID-19 community prevalence and shelter characteristics (e.g. ventilation, resident density, population turnover, and mask-wearing) determine prevalence in shelters [10]. Frequent symptom screening for COVID-19 is insufficient to prevent outbreaks due to pre- or asymptomatic infectious individuals [10, 11]. Restructuring to allow physical distancing, testing and isolation, can decrease outbreaks [12]. The ability to interrupt transmission chains through early identification of infectious individuals by molecular tests depends on frequency of testing and speed of reporting [13, 14]. Infection transmission models for shelters, found that in addition to standard infection prevention methods, twice-weekly testing may be necessary, and sometimes insufficient, to decrease viral reproduction numbers [15–18].

Nucleic acid amplification tests (e.g. real-time Reverse Transcription Polymerase Chain Reaction, rRT-PCR) are sensitive and specific, but they are expensive, require specialized equipment and personnel, and have disclosure delays that may facilitate transmission [19]. Antigen-based tests offer faster turnaround times (15–30 minutes), lower costs, and less specialized training [20]. Testing in moderate prevalence areas using Abbott's BinaxNOW COVID-19 Ag Card ("BinaxNOW") antigen test demonstrated high sensitivity and specificity for identifying infections with a transmissible viral load [21]. Along with referrals to isolation and quarantine settings (I&Q) [22], rapid testing and response can interrupt transmission and prevent outbreaks in congregate shelters.

Little is known about the implementation of rapid testing and response to COVID-19 infections in congregate shelters. Implementation frameworks, such as RE-AIM (Reach, Effectiveness, Adoption, Implementation, Maintenance), can aid in planning, adapting and evaluating interventions for implementation and dissemination [23]. We describe the BinaxNOW Shelter Pilot, a voluntary rapid, twice-weekly testing protocol for congregate shelters in San Francisco, California during January and February 2021. We apply the RE-AIM framework (Reach, Effectiveness, Adoption, Implementation and Maintenance) to evaluate the uptake and effectiveness of the BinaxNOW Shelter Pilot and inform recommendations to scale up similar protocols.

## Methods

### Study setting and design

The UCSF Benioff Homelessness and Housing Initiative (BHHI), the San Francisco Department of Public Health (SFDPH) and Department of Homelessness and Supportive Housing (SFHSH) developed and implemented the BinaxNOW shelter pilot. Initially, we implemented the pilot at four shelters and added six shelters during the study.

All residents and staff in adult, transitional age youth, and family shelters with congregate dormitory settings in San Francisco were eligible. To detect and isolate individuals as close to their infective period as possible, given resource limitations, test sensitivity, and disease progression (time between exposure, infection, and test positivity), we offered participants twice-weekly testing (Monday/Thursday or Tuesday/Friday) during the study period of January 14[th] to February 19[th], 2021.

All participants (or parents or guardians for those under 18) provided verbal informed consent. Consent was documented by non-clinical or clinical staff in a web-based data management platform (PrimaryHealth). We collected data required by state guidelines (contact information, race/ethnicity, gender, and date of birth) in PrimaryHealth. The UCSF Committee on Human Research (UCSF's Institutional Review Board) stated that the project did not meet criteria for human subjects research (and thereby did not require formal Institutional Review Board review) because this study was an evaluation of a public health intervention in the context of "public health surveillance" for SARS-CoV2, rather than a research study.

### BinaxNOW shelter pilot implementation

We based our protocol on previous test-and-respond workflows (Table 1), which involved monthly rRT-PCR for shelter residents and encouragement of shelter staff to seek community testing twice monthly. An outbreak team worked with shelter leadership to refer positive individuals to I&Q and conduct a case investigation and contact tracing.

**Staffing and sites preparation.**    A detailed description of staffing and supplies is available online [24]. We recruited "shelter champions," shelter staff members whom we trained and provided with a stipend to advertise testing events, coordinate event setup, and facilitate resident and staff participation. BHHI hired non-clinical staff and laboratory technicians, and recruited community volunteers; SFDPH provided registered nurses (RN) and health workers.

**On-site workflow.**    Initially, we had four testing stations with 8–10 staff (Fig 1): (1) Check-In: three to four non-clinical staff checked in participants, performed a symptom screen [25], provided information about isolation and quarantine, and labeled testing materials. A RN assessed individuals who reported symptoms consistent with COVID-19 to determine if they required isolation; (2) Swabbing: one to two technicians performed anterior nares swab [26]; (3) Testing: one BinaxNOW-trained staff (Tester) started the reaction [26], and one (Recorder) monitored the reaction time, read the result with the Tester, and recorded results on Primary-Health. The RN served as tiebreaker when there was disagreement. (4) Results: A non-clinical staff disclosed negative results to participants [24].

We performed rRT-PCR tests for the first 40 participants for validation [27].

**Result disclosure and BinaxNOW-positive participant investigation.**    The RN, with the shelter champion, located BinaxNOW-positive residents, isolated them, and conducted a case investigation and contact tracing. When staff tested positive, the RN phoned them, gave them isolation guidelines, and elicited if they had close contacts with residents while infectious. We referred close contacts to I&Q.

**Table 1. Workflow comparisons pre- and post-implementation of rapid SARS-CoV2 antigen testing at homeless shelters, San Francisco, 2021.**

| PRE-IMPLEMENTATION OF ANTIGEN TESTING (BinaxNOW) WORKFLOW | POST-IMPLEMENTATION OF ANTIGEN TESTING (BinaxNOW SHELTER PILOT) WORKFLOW |
|---|---|
| *SYMPTOM SCREENING* | |
| • Non-clinical shelter staff screen residents for symptoms daily and on entry to shelter<br>• Screening increases to twice daily when a case/outbreak is identified | • Screening guidance remains the same<br>• Clinical and non-clinical BinaxNOW team staff perform additional symptom screening of all participants during each testing event. |
| *COVID-19 EDUCATION & ENGAGEMENT* | |
| • Days of outreach and COVID-19 education in advance of every testing event. | • Frequent COVID-19 education and engagement during testing events. |
| *SCREENING TESTING* | |
| **Residents**:<br>• Monthly screening rRT-PCR tests provided at 5 of 12 shelters; no testing at others<br>• Reliant on updated shelter rosters, resident demographics and placing individual orders into electronic medical record system at every event. (hours)<br>• Result in 24–72 hours<br>• Outbreak team follows and discloses results (days)<br>**Staff**:<br>• Encouraged to test outside of work, twice monthly. Results managed independently of the outbreak team | • Twice-weekly BinaxNOW tests done at 10 of 12 shelters available to all residents and staff<br>• Reliant on PrimaryHealth for one-time registration (minutes per participant) and subsequent check-in (minutes per participant)<br>• Results in 30 minutes<br>• BinaxNOW team discloses results to staff and residents (minutes)<br>• BinaxNOW team notifies Outbreak team during event<br>• Confirmatory nucleic acid amplification testing of asymptomatic BinaxNOW-positive and symptomatic BinaxNOW-negative participants (Results in 24–72 hours) |
| *ISOLATION* | |
| • Outbreak team provides isolation guidance and refers individuals to I&Q OR supports shelter staff to refer individuals to I&Q (hours to day) | • Symptomatic BinaxNOW-negative and all BinaxNOW-positive participants are given isolation guidance. Residents are referred to I&Q during testing events (minutes to hours) |
| *CASE INTERVIEW & CONTACT TRACING* | |
| • Case interview and contact tracing performed after individual arrives at I&Q (hours to days)<br>• Outbreak team notifies close contacts and provides support quarantine guidance/ I&Q referrals (hours to days) | • Case interview and contact tracing done during testing events<br>• Resident close contacts notified and referred to I&Q during the testing event (minutes to hours) |
| *CASE RESPONSIVE TESTING* | |
| • Testing request submitted to Outbreak team for weekly rRT- PCR testing of both staff and residents<br>• Testing scheduled 24–72 hours later | • All residents and staff notified of potential exposure, encouraged to continue testing twice weekly |

We provided results within the same shift. PrimaryHealth sent automated emails and text messages to all participants with a link to their results. They sent reports to state and local health departments.

**I&Q referral, confirmatory testing and outbreak team integration.** We defined outbreaks as at least three COVID-19 cases within a 14-day period in epidemiologically-linked residents and/or staff (i.e., persons with a close contact to a case or a member of risk cohort) [28].

We isolated symptomatic BinaxNOW-negative and all BinaxNOW-positive participants. The RN or shelter champion completed I&Q referrals. Upon arrival to I&Q, asymptomatic BinaxNOW-positive and symptomatic BinaxNOW-negative individuals received confirmatory tests (rRT-PCR or transcription mediated amplification, TMA). Confirmatory test-negative residents returned to the shelter.

Our team notified the shelter of any positive cases, who paused new admissions upon identification of two BinaxNOW-positive cases or one BinaxNOW-positive and one symptomatic BinaxNOW-negative case in a 14-day period.

## Initial Worflow

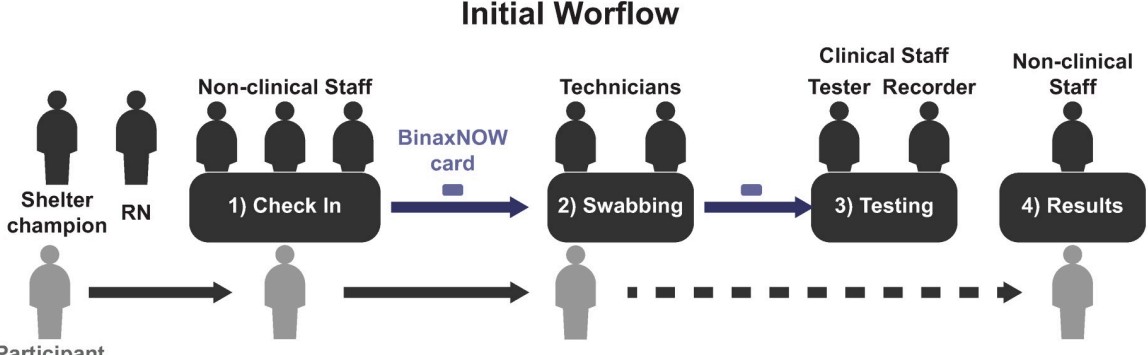

## Worflow after consolidation

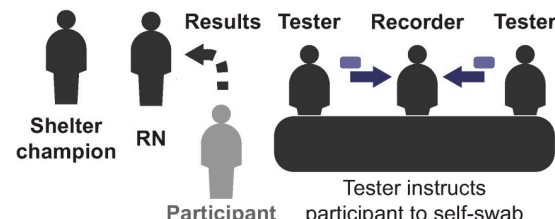

**Fig 1. Initial and post-consolidation of rapid SARS-CoV2 antigen testing at homeless shelters, San Francisco, 2021.** Initial BinaxNOW Shelter Pilot workflow: non-clinical staff checked in participants and labeled testing materials (1). Laboratory technicians then swabbed the participants (2) and handed the sample to a testing team (3) who ran the assay. Participants left the testing area and could return to get their results from a non-clinical staff (4). The shelter champion helped recruiting participants and the registered nurse (RN) assessed symptoms and disclosed positive results and conducted case investigation. BinaxNOW Shelter Pilot workflow after consolidation: To streamline the process and reduce staff and supplies needed, Testers registered, checked-in, instructed participants to self-swab and handed assays to the Recorder. The shelter champion helped recruiting participants and the registered nurse (RN) assessed symptoms and disclosed positive results and conducted case investigation.

**Resident participation incentives.** We provided monetary incentives to residents who tested regularly (twice in the first two weeks of testing or >75% of tests over the last four weeks) in all but one shelter.

## Iteration: Protocol modifications

Team members met daily on weekdays to discuss protocol modifications.

**Modifications to staffing and supplies preparation.** Starting in week three, to improve the implementation of the pilot, we conducted online training sessions for shelter champions (COVID-19 basics and role overview).

**Modifications to on-site workflow.** Starting in week five, to increase the number of shelters testing and enable maintenance without external (BHHI) resources, we reduced steps, supplies and staff [24]. We incorporated self-swabbing and reduced teams to five personnel: two Testers who, in addition to their original role, checked-in participants, instructed them on self-swabbing, and supervised self-swabbing; one Recorder; and one RN who, in addition to their original role, disclosed negative results. The shelter staff champion continued in their role. To optimize flow of patients and samples, we arranged the team in one station (Fig 1).

The Testers gave self-swabbing instructions at the time of collection. We provided no other prior training to participants. The participants immediately handed the swabs to the Tester after sampling to start the reaction.

To try to increase reach, we modified schedules.

**Modifications to result disclosure and BinaxNOW-positive participant investigation.** In week six, in response to miscommunication about a positive case, we modified the protocol to ensure case identification: (1) the RN reviewed staff responsibilities, and all symptomatic BinaxNOW-negative and BinaxNOW-positive participants at each event; and (2) Primary-Health sent a report with symptomatic BinaxNOW-negative and all BinaxNOW-positive participants to the team every day.

**Modifications to I&Q referral, confirmatory testing and outbreak team integration.** Starting in week three, because not all asymptomatic BinaxNOW-positive residents received confirmatory testing at the I&Q hotels, the RN completed all I&Q referrals and assessed adherence to confirmatory testing.

## Implementation evaluation

**Overall approach.** We used the RE-AIM framework to guide implementation planning and evaluation (Table 2). We established reach and effectiveness criteria to mirror shelter realities (reach measures). We evaluated case detection and outbreak control (effectiveness). For adoption, we focused on the number of shelters participating, and their ability to offer testing to eligible participants. For implementation, we assessed the consistency of protocols and adaptations made to increase reach and effectiveness. For maintenance, we assessed whether SFDPH was able to sustain the project without outside resources.

**Data collection.** Reach: SFHSH provided de-identified individual resident census data (including demographics, date of birth, and admission and discharge dates), and staff census data. To assess how many residents were on-site during testing events, the testing team conducted a headcount within the facilities during testing events. To estimate staff who were on-site during daytime testing events, SFHSH provided a count of all staff working day shifts.

Effectiveness: We recorded symptoms and test results on PrimaryHealth. We obtained confirmatory test results from the electronic health record.

We also recorded the timeframe of implementation and number of shelters where we implemented BinaxNOW Testing (adoption), the rationale for changes to protocols, and their outcomes (implementation), and the number of shelters where SFDPH offered testing after the study period (maintenance).

**Data analysis.** We analyzed data using Matlab [29].

We used census data for the study period corresponding to each shelter. We excluded residents admitted and discharged on the same day. We conducted Mann-Whitney U test to compare age, and chi-squared test to compare gender for the participating and non-participating resident populations. We included shelters with <10% missing gender data in gender comparisons based on the average percent and confidence interval across shelters. We were unable to examine race and ethnicity due to incomplete data.

To determine the period participant residents lived in a shelter, we matched them to the shelter census database using their date of birth. We used the mean number of residents on-site. We assumed participant staff had continuous employment over the study period.

To estimate adherence to twice-weekly testing, we calculated the proportion of tests out of the total number of testing events offered per shelter. We included only shelters with at least six testing events. Because some shelters had high resident turnover, we included only those individuals who remained in the shelter throughout the shelter's testing period.

**Table 2. RE-AIM framework to evaluate the implementation of rapid SARS-CoV2 antigen testing in homeless shelters, San Francisco, 2021.**

| | Step | Level | Question | Source | Measure |
|---|---|---|---|---|---|
| **Reach**<br>the number, proportion, and representativeness of individuals (and sites) who are willing to participate in BinaxNOW Shelter Pilot | 1<br>Identification and characterization of total eligible population | Shelter | What is the number of eligible shelters? | SFHSH[a] database | Number of shelters |
| | | Residents | How many residents live in eligible shelters, and what are their demographics? | SFHSH[a] database | Total number of beds per shelter |
| | | | | Shelters' resident database | Total number of residents per shelter<br>Demographics of residents per shelter |
| | | Staff | How many staff work in eligible shelters, and what are their demographics? | Shelters' staff database | Total number of staff per shelter<br>Demographics of staff per shelter |
| | 2<br>Identification and characterization of on-site sample during testing period | Residents | How many residents stayed in the shelter during the study, and how transient was this population? | Shelters' resident census | Residents length of stay in the shelter |
| | | | How many residents were present on-site during testing events? | Study headcount | Proportion of residents present in the shelter at time of testing |
| | | Staff | How many staff were present on-site during testing events? | Shelters' staff shift data | Proportion of staff present in the shelter at time of testing |
| | 3<br>Completion of twice-weekly testing | Resident and Staff | What proportion of residents and staff participated in testing? And what proportion of the present-during-event population participated in testing? | PrimaryHealth | Proportion of residents and staff that tested at least once |
| | | Resident and Staff | Did residents and staff adhere to twice-weekly frequency of testing? | PrimaryHealth | Frequency of testing and proportion of residents and staff who participated in 100% of tests offered to them |
| | | Resident and Staff | Was the participating testing population representative of the shelter population? | Shelters' staff and resident databases and PrimaryHealth | Demographics of residents and staff who tested vs total population |
| **Effectiveness**<br>the impact of rapid on-site testing on key individual outcomes, including assessment of adverse effects | 4<br>Identification of positives and effective isolation | Resident and Staff | How many BinaxNOW-negative symptomatic participants did we detect by twice-weekly testing during the study period? | PrimaryHealth | Number of BinaxNOW-negative symptomatic participants |
| | | Resident and Staff | What proportion of BinaxNOW-negative PUIs tested positive on confirmatory tests? | Health Record | Number of positive confirmatory results in BinaxNOW-negative subpopulation |
| | | Resident and Staff | How many BinaxNOW-positive did we detect by twice-weekly testing during the study period? | PrimaryHealth | Number of BinaxNOW-positive participants |
| | | Resident and Staff | Did the intervention lead to immediate (during the testing event) isolation of BinaxNOW-positive participants regardless of symptoms or symptomatic BinaxNOW-negative participants? | PrimaryHealth | Number of symptomatic BinaxNOW-negative and all BinaxNOW-positive participants identified and isolated during the event |
| | 5<br>Identification of incongruent confirmatory tests | Shelter, Resident and Staff | How many BinaxNOW-positive cases tested negative in confirmatory tests? | PrimaryHealth | Number of BinaxNOW-positive participants referred to isolation and quarantine who tested negative on confirmatory test |
| | 6<br>Identification of outbreaks | Shelter | How many outbreaks were detected? And what proportion were resolved within 28 days from the last positive case? | PrimaryHealth | Number of outbreaks |
| | | Shelter | How many cases did not develop into outbreaks? | PrimaryHealth | Number of shelters with isolated cases |

*(Continued)*

**Table 2.** (Continued)

|  | Step | Level | Question | Source | Measure |
|---|---|---|---|---|---|
| **Adoption**<br>the number, proportion, and representativeness of settings and intervention agents (people who deliver the program) for delivery of the rapid on-site testing intervention | 7<br>Shelter-level offering of testing | SFDPH[b] and Shelter | How many sites adopted BinaxNOW Shelter Pilot in the study period? | PrimaryHealth | Number of shelters over time |
|  |  | SFDPH[b] and Shelter | How promptly was twice-weekly testing implemented? | Notes from daily team meetings | Time delay from contacting to first testing event |
|  |  | SFDPH[b] and Shelter | Did shelters offer twice-weekly testing to all eligible participants? | Notes from daily team meetings/ testing calendar | Number of shelters offering tests to staff and residents over time |
| **Implementation**<br>at the setting level, the fidelity to core components of the on-site rapid testing protocols for, and the types of adaptations made to accommodate important variation in site operations | 8<br>Adaptation and fidelity of implementation | SFDPH[b] and Shelter | Were operations faithfully reproduced in multiple shelters and by different teams? | Notes from daily team meetings | Number of shelters offering tests following BinaxNOW Shelter Pilot workflows |
|  |  | SFDPH[b] and Shelter | Did testing teams and shelter leadership integrate adaptations to increase Reach, and did they increase Reach? | Notes from daily team meetings | Number of adaptations assimilated by SFDPH and shelter leadership<br>Number of participants before and after adaptations were implemented |
|  |  | SFDPH[b] and Shelter | Did testing teams and shelter leadership integrate adaptations to enable Adoption? | Notes from daily team meetings | Number of adaptations assimilated by SFDPH[b] and shelter leadership |
| **Maintenance**<br>the extent to which sites maintained elements of the on-site rapid testing program (or intervention) after the study period, and to what extent the program was expanded to additional sites | 9<br>Transference of BinaxNOW Testing to SFDPH[b] | SFDPH[b] | Was SFDPH[b] able to sustain the program without external support? | Records from SFDPH[b]; notes from meetings | Proportion of enrolled shelters where SFDPH[b] conducted testing after the end of study period<br>Resources provided by BHHI[c] after the end of the study period |
|  |  | SFDPH[b] | Was SFDPH[b] able to expand the program? | Records from SFDPH[b]; notes from meetings | Number of shelters enrolled by SFDPH[b] after the end of study period |

[a]San Francisco Department of Homelessness and Supportive Housing.

[b]San Francisco Department of Public Health.

[c]UCSF Benioff Homelessness and Housing Initiative.

All reported ranges refer to the range of values across shelters.

## Results

### Reach

**Identification and characterization of total eligible participants.** We implemented the pilot at ten of 12 eligible shelters during the study period. During the study, 828 unique residents lived in and 435 staff worked at these ten shelters. The mean age among residents and staff was 44 years. Less than one third (28.8%) of residents and nearly half (49.8%) of staff were cis women (Table 3).

**Identification and characterization of on-site sample during testing period.** On average, the daily census included 645 residents. Over half (57.9% n = 636 in 6 shelters) of residents remained in shelters for the duration of the study (Table 3). Less than half of residents (45.2%, range 10.9% to 81.9%) and one third of staff (31.3%, range 5.4% to 68%) were on-site during testing events (Table 4).

**Completion of twice-weekly testing.** Half of eligible residents (47.5%, range 16.5% to 81.4%) and shelter staff (53.3%, range 9.6% to 128%—the latter meaning some staff off their shift tested) participated in testing at least once. On average, one-quarter (24.1%, range 7.3% to 66.7%) of residents and one-fifth (19.6%, range 5.8% to 45.8%) of the staff participated in

**Table 3. Characteristics of shelter residents and staff during implementation of rapid SARS-CoV2 antigen testing at homeless shelters, San Francisco, 2021.**

| | Residents | | Staff | |
|---|---|---|---|---|
| | Eligible | Tested at least once (% of eligible) | Eligible | Tested at least once (% of eligible) |
| **N** | 828 | 393 (47.5%) | 435 | 232 (53.3%) |
| **Age (years)** | | | | |
| Mean Age [95% CI] | 44 [43.2–44.9] | 45.8 [44.5–47.1] | 44 [N/A] | 44.9 [43.1–46.7] |
| 0–17 | 10 (1.2%) | 6 (1.5%) | N/A | 0 (0%) |
| 18–24 | 39 (4.7%) | 21 (5.3%) | N/A | 18 (7.8%) |
| 25–49 | 482 (58.2%) | 194 (49.4%) | N/A | 114 (49.1%) |
| 50–64 | 254 (30.7%) | 145 (36.9%) | N/A | 84 (36.2%) |
| 65+ | 42 (5.1%) | 27 (6.9%) | N/A | 16 (6.9%) |
| Not available | 1 (0.1%) | 0 (0%) | N/A | 0 (0%) |
| **Gender** | | | | |
| Female | 147 (28.8%) | 79 (30.3%) | 134 (49.8%) | 65 (56.0%) |
| Male | 329 (64.4%) | 175 (67.0%) | 135 (50.2%) | 48 (41.4%) |
| Transgender / Genderqueer/ Gender Non-binary | 2 (0.4%) | 1 (0.4%) | 0 (0%) | 2 (1.7%) |
| Unavailable | 33 (6.5%) | 6 (2.3%) | 0 (0%) | 1 (0.9%) |

each event. More than half of the residents (53.2%, range 34.0% to 81.9%) and dayshift staff (62.8%, range 17.6% to 255%) on-site participated per event (Table 4, S1 File). We found no significant demographic differences between participant and non-participant or eligible populations (S1 Appendix).

Residents completed on average 36.8% and staff 34.1% of the tests offered in the shelters where we tested at least three weeks during the study period (n = 196 for residents, n = 198 for staff). Among these participants, most tested only once; only 4.1% of residents and 3.0% of staff tested twice-weekly during the study (i.e., 100% adherence, Fig 2A, S1 File). Participation was stable over time, although two shelters showed a noticeable increase (shelter 2) or decrease (shelter 5) in participation (Fig 2B).

## Effectiveness

**Identification of positives and effective isolation.** We performed concomitant rRT-PCR on the first 40 participants, all of whom were asymptomatic and BinaxNOW negative. Two

**Table 4. Participation in shelters included in the pilot implementation of rapid SARS-CoV2 antigen testing, San Francisco, 2021.**

| Shelter | 1 | 2 | 3 | 4 | 5 | 6 | 7 | 8 | 9 | 10 | Total |
|---|---|---|---|---|---|---|---|---|---|---|---|
| Total testing events | 11 | 11 | 10 | 8 | 8 | 6 | 4 | 3 | 2 | 2 | |
| **Residents** | | | | | | | | | | | |
| Relative population size (normalized to largest shelter, n = 142) | 0.44 | 0.46 | 1.00 | 0.32 | 0.64 | 0.47 | 0.16 | 0.55 | 0.36 | 0.13 | |
| Presence (% of residents in census) | 38.4 | 41.7 | 56.1 | 31.1 | 39.3 | 41.8 | 70.4 | 10.9 | 81.4 | 86 | 45.2 |
| Residents discharged or admitted during study (% of total residents) | 22.9 | 84.7 | 30.8 | 27.5 | 20.4 | 41.1 | 26.9 | 27.5 | 3.8 | 34.8 | 37.4 |
| Participation (average % of participants from census) | 28.3 | 9.2 | 17.7 | 12.3 | 12.3 | 18 | 21.2 | 6.2 | 65.4 | 47.8 | 18.8 |
| Participation (average % of participants from population present at time of testing) | 81.9 | 49.9 | 38.3 | 44.6 | 35.3 | 57.7 | 34 | 66.7 | 81.9 | 67.3 | 53.2 |
| **Staff** | | | | | | | | | | | |
| Relative size (normalized to largest shelter, n = 72) | 0.57 | 0.35 | 1.00 | 0.39 | 0.97 | 0.82 | 0.28 | 0.78 | 0.72 | 0.17 | |
| Dayshift staff (% of staff in census) | 22 | 68 | 25 | 35.7 | 37.1 | 47.5 | 25 | 5.4 | 32.7 | 25 | 31.3 |
| Participation (average % of participants from census) | 70.7 | 128 | 50 | 67.9 | 60 | 67.8 | 20 | 26.8 | 9.6 | 83.3 | 53.3 |
| Participation (average % of participants from dayshift staff) | 114 | 65.8 | 68.3 | 56.3 | 41.3 | 61.3 | 40 | 256 | 17.6 | 183 | 62.8 |

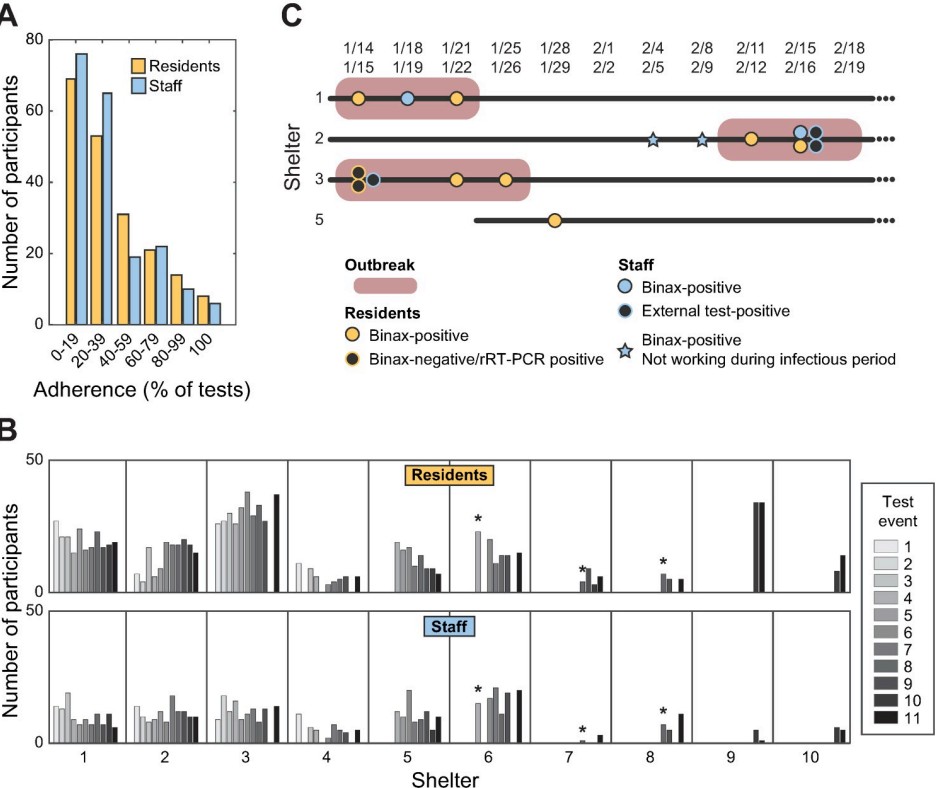

**Fig 2. Participation in rapid SARS-CoV2 antigen testing and positive cases over time at homeless shelters, San Francisco, 2021.** A) Histogram of adherence (percentage of tests taken by a participant from tests available to them; n = 394, within 6 shelters with 6 or more testing events). B) Number of participants over testing events per shelter. Missing bars indicate no testing event in that date. Shelters 5–10 were added to the BinaxNOW Testing Pilot after the start of the study period. *: Scheduled testing times were modified after indicated event. C) 3 outbreaks and 1 isolated BinaxNOW-positive case were identified. All outbreaks were resolved. No cases were identified in the other 6 shelters during the study period.

(5%) residents tested positive on rRT-PCR. Follow-up tests three days later in I&Q found that the first resident tested negative on BinaxNOW and rRT-PCR, and the second participant, who subsequently developed symptoms, tested positive on both tests.

BinaxNOW prevalence was 2.2%: ten residents (2.5%), and four staff (1.8%) in four different shelters tested positive (Fig 2C). Of the 14 BinaxNOW-positive participants, only one resident and two staff were symptomatic. We referred eight of ten residents to I&Q on the same date of the event. Of the two not referred, one resident received isolation guidance and left the shelter. The second resident was not isolated until five days after their positive test due to miscommunication. We identified no resident close contacts during case investigations. The float RN contacted all BinaxNOW-positive staff.

One symptomatic BinaxNOW-negative resident was referred to I&Q. The confirmatory test was negative, and they returned to the shelter.

No residents received a COVID-19 diagnosis outside of BinaxNOW testing during the study; three staff members did.

**Identification of incongruent confirmatory tests.** Of the eight asymptomatic Binax-NOW-positive residents referred to I&Q, three were confirmed positive; two did not receive a confirmatory test; and three tested negative on confirmatory testing. Two of the three BinaxNOW-positive/confirmatory-negative residents were BinaxNOW-positive in

subsequent testing events and were referred to I&Q again, where they tested negative by confirmatory tests.

**Identification of cases and outbreaks.** We identified cases in four of the ten shelters. Three shelters met criteria for outbreaks (i.e., ≥ three cases) but no outbreak exceeded five cases (Fig 2C). The three shelters with outbreaks were large, had transient resident populations, or had high participation (Table 4). One shelter had one isolated positive case.

## Adoption

**Shelter-level offering of BinaxNOW shelter pilot.** We implemented testing in ten shelters within weeks of initiating planning. We delayed staff testing for one week for staff to complete release forms.

## Implementation

**Adaptation and fidelity of implementation.** The testing team adopted the new workflow (Fig 1) in all shelters within a week of trying it in one shelter. These changes led to a significant decrease in staffing and supply needs.

We did not see an increase in participation after changes in scheduled testing times due to shelter champion input (Fig 2B).

We replicated the same workflow and protocol modifications in all shelters with few modifications (e.g., incentive structure).

We conducted 1,142 tests before self-swabbing implementation, of which 8 were positive and 4 were false positive (0.70% true positivity). We conducted 562 tests by self-swabbing, of which 4 were positive and 1 was a false positive (0.71% true positivity).

The testing team and shelter champions adopted changes aiming to address deviations to the protocols (e.g., missing confirmatory testing at I&Q or delaying positive case disclosure). We completed all resident confirmatory tests after week three and contacted all BinaxNOW-positive during the event after week six.

## Maintenance

**Transference of BinaxNOW testing to SFDPH.** SFDPH was able to sustain testing in all ten shelters and expanded to all eligible shelters with limited support from BHHI (two non-clinical staff and PrimaryHealth contract) after the end of the study period.

## Discussion

In a pilot twice-weekly BinaxNOW antigen testing in congregate homeless shelters in San Francisco, we were able to implement testing, detect SARS-CoV-2 infections, isolate individuals, and identify outbreaks.

The BinaxNOW testing pilot reached approximately one quarter of eligible residents and staff, and one-half of those on-site. Limited reach could have resulted from residents leaving the shelter (residents had to check in every 48 hours to retain their beds). Participation levels varied largely across shelters, and in some shelters over time. This may have been due to shelter culture, the involvement level of shelter champions, or unmeasured characteristics of the residents. In some shelters, testing drew non-shift staff for testing.

Although approximately half of residents and staff tested at least once, there was a low adherence to twice-weekly testing. This may be due to our inability to convey the rationale behind twice weekly testing, discomfort with testing, a recognition of the implications of low community prevalence, or competing priorities. Other incentive structures (e.g., incentives at

each event or non-monetary incentives instead of delayed financial rewards) may be more motivating. Twice-weekly cadence may be unrealistic for some shelters. Models that indicate preventive effects depend on twice weekly testing may overestimate the true effectiveness of testing programs.

The effective reach of the BinaxNOW shelter pilot was higher than with prior testing regimens (five shelters once a month with an average uptake of 35% of residents (and no staff) at each event; S. Strieff, personal communication, March 22[nd], 2021). Due to rapid turnaround, we reduced delays in isolation of infectious individuals compared to previous workflows (Table 1). These responses may have contributed to preventing large outbreaks, despite a relatively narrow reach of testing.

Despite mid to low community prevalence (one-week average 30.9 and 7.7 new cases per day per 100,000 during the first and last week of the study, respectively) [30], we detected three outbreaks and one isolated case. The largest had five cases, fewer than prior outbreaks [11, 16, 31, 32]. This may have been a result of our intervention or other changes instituted since early in the pandemic (reduced capacity and improved adherence to mask-wearing) [33]. We detected most cases in asymptomatic participants, highlighting the limitations of symptom screening for isolation of infectious individuals [16]. While BinaxNOW is less sensitive than nucleic acid amplification tests, it appears effective at detecting infectious individuals [27]. Detecting asymptomatic individuals is key to interrupting transmission chains as asymptomatic individuals may account for more than half of all transmission [34].

We detected two of 40 individuals who were BinaxNOW-negative but rRT-PCR-positive. However, one of these was consistent with a prior infection [35], leading to unnecessary isolation, a potential problem with highly sensitive RT-PCR testing. The other individual's test patterns align with early infection, prior to high infectivity. High adherence to twice-weekly testing would be needed to detect these early cases.

We found a relatively high false positive rate, consistent with concerns of using BinaxNOW in low community prevalence and among asymptomatic persons [36]. Three of eight asymptomatic individuals were BinaxNOW-positive/confirmatory-negative, suggesting false BinaxNOW positives. This is costly in terms of transportation and I&Q requirements and may have undermined trust in testing and the healthcare system, raising questions about BinaxNOW testing during periods of low community prevalence. We recommend continuous dialogue and education of participants on BinaxNOW testing strengths and limitations.

As a result of the strong collaboration between academic, public health and homeless service providers, we were able to implement and adapt the pilot, allowing us to minimize errors, increase adoption and reduce resources, which contributed to maintenance. Self-swabbing had no noticeable effect on positivity rates or the ability to detect outbreaks. The SFDPH offered twice-weekly testing in all eligible shelters with little external support after the study, suggesting maintenance of this intervention. We found the web-based test reporting system (PrimaryHealth) to be instrumental to conduct mandatory reporting to local and state officials, and track data. Despite our efforts to streamline staff and resources, BinaxNOW screening testing still requires significant staff and resources. Further implementation studies should evaluate modifications to reduce workflow and staff and consider further task-shifting to nonclinical partners.

The study has several limitations. We focused on implementation of BinaxNOW testing rather than comparing different strategies. Thus, the contribution of twice-weekly testing to preventing and resolving outbreaks is unclear. While infection transmission models suggest that twice-weekly testing is necessary [16–18], we tested a lower proportion of eligible participants than recommended by models [18]. However, this may reflect the reality of testing regimens at homeless shelters, where residents do not tend to stay during the days. Those who

leave during the day could be at higher risk of contracting SARS-COV-2. We did not study the causes (e.g., low interest, previous COVID-19 infections, testing fatigue) and consequences (e.g., transmission) of low participation. We estimated shelter residents present by headcounts for three weeks; these may not be accurate. We had limited demographic data, which limited our ability to detect differences. In addition, BinaxNOW sensitivity is lower than nucleic acid amplification methods [21], implying that we could have missed infections within the participant population. Furthermore, we conducted our study for six weeks, in one jurisdiction which had implemented aggressive COVID prevention protocols in shelters (including reduced capacity and universal masking) during a period of low to moderate COVID-19 community incidence that preceded vaccine rollouts. Other settings and contexts could have different results. However, the lack of large-scale outbreaks is reassuring.

As the COVID-19 pandemic evolves, the trade-offs of frequent testing in congregate shelters will change. Effectiveness (and cost-effectiveness) of rapid testing for isolation and outbreak prevention should thus be evaluated in the specific setting context. Fluctuations in community transmission affect false positive and negative rates, and determine the probability of infection for individuals. Vaccination campaigns and other interventions may require a change in resource allocation. Vaccinations, as well as previous exposure, can affect the susceptibility of individuals to severe illness and change the population at risk. The rise of new variants has the potential of changing the epidemiology of the disease, as well as the sensitivity of diagnostic tests. While we expect rapid test and response programs to be less cost-effective with high vaccination rates and low community transmission, all these factors should be considered when implementing measures to prevent outbreaks in congregate settings.

Our pilot allowed for detection and isolation of COVID-19 cases among staff and residents. This intervention could be applied to other congregate settings where there is a high turnover of residents, a higher risk of transmission, and challenges to physical distancing [37]. Lessons learned from our rapid test and respond model, and application of an implementation framework matching the cascade of care could be applied to how we approach detection and response for other infectious diseases in this population with additional considerations of how to assign staff and resources to needs. The partnership between an academic group, and representation from city departments that manage public health and the shelters, with additional training and support for a shelter champion could be applied to vaccination or other public health efforts. Low-barrier frequent testing with rapid turnaround is an important public health intervention to improve accessibility and availability of testing and rapid identification and isolation of COVID-19 positive cases in shelters.

## Supporting information

**S1 Appendix. Demographic description of participant and eligible populations.** (DOCX)

**S1 File. Census and participant number.** Individual shelter participation per event and total census population. This data is shown in Fig 2B. Percentage of tests taken. Individual datapoints shown in Fig 2A. (XLSX)

## Acknowledgments

Sandra Nicholson, Lunden Stiggers, Rocio Novoa, Cesar Cardenas, Gabriela Alvarez, Jorge Contreras, Celeste Enriquez, Tamar Schnepp, Lena Simbe and Princess Luna provided on the ground input for the implementation of this project. Jennifer Coffey, Louis Bracco, Lisa

Rachowitz, Scott Walton, Howard Chen, Elisabet Medina, Ronald Thai, Ashley Scarborough, facilitated data availability, coordinated work with multiple stakeholders and gave feedback throughout. Carina Marquez, Genay Pilarowski and Jackie Martinez provided technical advice and support. This project would not have been possible without the help of shelter leadership, especially the shelter champions, and community volunteers.

## Author Contributions

**Conceptualization:** Andrés Aranda-Díaz, Elizabeth Imbert, Sarah Strieff, Margaret A. Handley, Margot Kushel.

**Data curation:** Andrés Aranda-Díaz, Sarah Strieff, Dave Graham-Squire, Jennifer L. Evans.

**Formal analysis:** Andrés Aranda-Díaz, Sarah Strieff.

**Investigation:** Andrés Aranda-Díaz, Elizabeth Imbert, Sarah Strieff.

**Methodology:** Andrés Aranda-Díaz, Elizabeth Imbert, Sarah Strieff, Jamie Moore, Margaret A. Handley, Margot Kushel.

**Supervision:** Willi McFarland, Jonathan Fuchs, Margaret A. Handley, Margot Kushel.

**Visualization:** Andrés Aranda-Díaz.

**Writing – original draft:** Andrés Aranda-Díaz, Elizabeth Imbert, Margaret A. Handley, Margot Kushel.

**Writing – review & editing:** Andrés Aranda-Díaz, Elizabeth Imbert, Sarah Strieff, Dave Graham-Squire, Jennifer L. Evans, Willi McFarland, Margot Kushel.

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
