## [Decision Letter · Decision Letter 0]

22 Dec 2021

PONE-D-21-13440Implementation of Rapid and Frequent SARS-CoV2 Antigen Testing and Response in Congregate Homeless SheltersPLOS ONE

Dear Dr. Aranda-Diaz,

Thank you for submitting your manuscript to PLOS ONE. After careful consideration, we feel that it has merit but does not fully meet PLOS ONE’s publication criteria as it currently stands. Therefore, we invite you to submit a revised version of the manuscript that addresses the points raised during the review process.

Please go through the reviewer comments Please submit your revised manuscript by Feb 05 2022 11:59PM. If you will need more time than this to complete your revisions, please reply to this message or contact the journal office at plosone@plos.org. Please include the following items when submitting your revised manuscript:A rebuttal letter that responds to each point raised by the academic editor and reviewer(s). You should upload this letter as a separate file labeled 'Response to Reviewers'.A marked-up copy of your manuscript that highlights changes made to the original version. You should upload this as a separate file labeled 'Revised Manuscript with Track Changes'.An unmarked version of your revised paper without tracked changes. You should upload this as a separate file labeled 'Manuscript'.

We look forward to receiving your revised manuscript.

Kind regards,

Prasenjit Mitra, MD, CBiol, MRSB, MIScT, FLS, FACSc, FAACC

Academic Editor

PLOS ONE

Journal Requirements:

"The project received funding from the UCSF Benioff Homelessness and Housing Initiative, Kaiser Community Benefits, and Heluna Health. "

Additional Editor Comments (if provided):

Please go through the reviewer comments and make necessary revisions.

Reviewers' comments:

Reviewer's Responses to Questions

**Comments to the Author**

1. Is the manuscript technically sound, and do the data support the conclusions?

Reviewer #1: Yes

Reviewer #2: Yes

Reviewer #3: Yes

2. Has the statistical analysis been performed appropriately and rigorously? 

Reviewer #1: Yes

Reviewer #2: Yes

Reviewer #3: Yes

3. Have the authors made all data underlying the findings in their manuscript fully available?

Reviewer #1: Yes

Reviewer #2: Yes

Reviewer #3: Yes

4. Is the manuscript presented in an intelligible fashion and written in standard English?

Reviewer #1: Yes

Reviewer #2: Yes

Reviewer #3: Yes

5. Review Comments to the Author

Reviewer #1: May describe in more detail about the ethical issues. What do you mean by 'public health surveillance' exemption?

In the discussion may consider the varying impact of the effectiveness of rapid and frequent testing depending on the stage of the pandemic. For instance in early stage of the pandemic with higher number of susceptible people this may be more effective and later in the pandemic when susceptible people are very few this measure may not be cost effective.

Reviewer #2: The study is a well planned out one and the RE-AIM model is implemented to an acceptable level. The data is explicitly presented in toto and the analysis leads to the expected objectives. The manuscript is written a simple understandable manner and gives a reference to the limitations of the study.

Reviewer #3: After the iteration processes, you have scaled up testing and advised study participants to do self-testing.

1.There is no mention of any appraisal training through online mode to them on self testing.

2.There is no mention of number of tests done by participants (self swab samples) & their results.

Authors should give seperate details of the number of these self swab tests and their reslts as they have bearing on the results of the study..

3.What was the time elapsed in the handover of self swab assays to the recorder?

4.Can you give us the rationale behind twice a week testing.?

6. PLOS authors have the option to publish the peer review history of their article (what does this mean?). If published, this will include your full peer review and any attached files.

Reviewer #1: **Yes: **Professor Amitav Banerjee

Reviewer #2: **Yes: **PRASANNA KAMATH B T

Reviewer #3: **Yes: **Khalid Bashir

---

## [Author Response · Author response to Decision Letter 0]

21 Jan 2022

Reviewer #1: 

May describe in more detail about the ethical issues. What do you mean by 'public health surveillance' exemption?

We have added language to the Ethics Statement in the Methods to clarify public health surveillance exemption. 

In the discussion may consider the varying impact of the effectiveness of rapid and frequent testing depending on the stage of the pandemic. For instance in early stage of the pandemic with higher number of susceptible people this may be more effective and later in the pandemic when susceptible people are very few this measure may not be cost effective.

We appreciate the reviewer’s comments. We have added a paragraph in the discussion listing factors that can contribute to the cost-effectiveness and effectiveness of rapid testing and response.

Reviewer #2: 

The study is a well planned out one and the RE-AIM model is implemented to an acceptable level. The data is explicitly presented in toto and the analysis leads to the expected objectives. The manuscript is written a simple understandable manner and gives a reference to the limitations of the study.

We appreciate the reviewer’s favorable comments!

Reviewer #3: 

After the iteration processes, you have scaled up testing and advised study participants to do self-testing.

1.There is no mention of any appraisal training through online mode to them on self testing.

As explained in a new paragraph under Modifications to On-site Workflow in the Methods, we provided no other training but the one offer by the Testers on-site.

2.There is no mention of number of tests done by participants (self swab samples) & their results.

Authors should give seperate details of the number of these self swab tests and their reslts as they have bearing on the results of the study..

A new paragraph has been added to the Implementation section in the Results. We have also added language to the discussion to highlight that we have no indication that self-swabbing altered the results.

3.What was the time elapsed in the handover of self swab assays to the recorder?

We have included language in the methods to clarify that the swabs were immediately handed to the Testers after sampling.

4.Can you give us the rationale behind twice a week testing.?

We have added language in the Study Setting and Design section to clarify this point.

---

## [Editor Report · Decision Letter 1]

21 Feb 2022

Implementation of Rapid and Frequent SARS-CoV2 Antigen Testing and Response in Congregate Homeless Shelters

PONE-D-21-13440R1

Dear Dr. Aranda-Diaz,

We’re pleased to inform you that your manuscript has been judged scientifically suitable for publication and will be formally accepted for publication once it meets all outstanding technical requirements.

Kind regards,

Prasenjit Mitra, MD, CBiol, MRSB, MIScT, FLS, FACSc, FAACC

Academic Editor

PLOS ONE
---

## [Editor Report · Acceptance letter]

1 Mar 2022

PONE-D-21-13440R1 

Implementation of Rapid and Frequent SARS-CoV2 Antigen Testing and Response in Congregate Homeless Shelters 

Dear Dr. Aranda-Diaz:

I'm pleased to inform you that your manuscript has been deemed suitable for publication in PLOS ONE. Congratulations! Your manuscript is now with our production department. 

Kind regards, 

on behalf of

Dr. Prasenjit Mitra 

Academic Editor

PLOS ONE